# Remodeling of Arterial Tone Regulation in Postnatal Development: Focus on Smooth Muscle Cell Potassium Channels

**DOI:** 10.3390/ijms22115413

**Published:** 2021-05-21

**Authors:** Anastasia A. Shvetsova, Dina K. Gaynullina, Olga S. Tarasova, Rudolf Schubert

**Affiliations:** 1Department of Human and Animal Physiology, Faculty of Biology, M.V. Lomonosov Moscow State University, 119234 Moscow, Russia; dina.gaynullina@gmail.com (D.K.G.); ost.msu@gmail.com (O.S.T.); 2Department of Physiology, Russian National Research Medical University, 117997 Moscow, Russia; 3Laboratory of Exercise Physiology, State Research Center of the Russian Federation-Institute for Biomedical Problems, Russian Academy of Sciences, 123007 Moscow, Russia; 4Physiology, Institute of Theoretical Medicine, Medical Faculty, University of Augsburg, 86159 Augsburg, Germany; rudolf.schubert@med.uni-augsburg.de

**Keywords:** potassium channels, postnatal development, vascular smooth muscle, systemic circulation, vascular remodeling

## Abstract

Maturation of the cardiovascular system is associated with crucial structural and functional remodeling. Thickening of the arterial wall, maturation of the sympathetic innervation, and switching of the mechanisms of arterial contraction from calcium-independent to calcium-dependent occur during postnatal development. All these processes promote an almost doubling of blood pressure from the moment of birth to reaching adulthood. This review focuses on the developmental alterations of potassium channels functioning as key smooth muscle membrane potential determinants and, consequently, vascular tone regulators. We present evidence that the pattern of potassium channel contribution to vascular control changes from K_ir_2, K_v_1, K_v_7 and TASK-1 channels to BK_Ca_ channels with maturation. The differences in the contribution of potassium channels to vasomotor tone at different stages of postnatal life should be considered in treatment strategies of cardiovascular diseases associated with potassium channel malfunction.

Potassium channels are transmembrane proteins which form potassium-selective pores. They are widely distributed among living organisms and are found in many cell types, including cells of the cardiovascular system. In cardiac and some vascular (mainly venous) myocytes, potassium channels mediate the repolarization phase of the action potential. In arterial smooth muscle cells, which commonly do not generate action potentials under physiological conditions, a change in potassium channel activity provides gradual shifts of the membrane potential [1,2]. This leads to alterations in the activity of voltage-gated calcium channels accompanied by changes in the calcium influx, as well as the intracellular calcium concentration and corresponding modifications of vascular tone. Of note, malfunctions of several potassium channel types are associated with arterial hypertension, diabetes mellitus and other metabolic disorders [3,4,5,6].

## 1. Potassium Channels Are Key Regulators of Arterial Tone

The membrane potential of arterial smooth muscle cells is a key determinant of vascular tone [7,8]. Indeed, a membrane potential shift by only a few millivolts towards hyperpolarization or depolarization leads to a considerable increase and decrease in arterial lumen, respectively [9]. Potassium channels of the cell membrane play a key role in the establishment of the resting membrane potential, as well as in the regulation of the membrane potential under the action of various vasoconstrictors and vasodilators [6,10]. Several functional types of potassium channels are described in arterial smooth muscle voltage-gated potassium channels (K_v_), inward-rectifier potassium channels (K_ir_), adenosine triphosphate-sensitive potassium channels (K_ATP_), two-pore potassium channels (K_2P_), as well as calcium-activated potassium channels of large conductance (BK_Ca_) [8,11].

It has been shown that potassium channel blockade leads to the depolarization of smooth muscle cells and an increase in contractile responses of the vascular wall to stimuli of different nature. Activation of potassium channels, on the contrary, counteracts the development of contraction or causes the dilation of arterial smooth muscle [8,12]. Therefore, potassium channels reduce the depolarization of arterial smooth muscle and play anticontractile or vasorelaxation roles. Importantly, potassium channel activity can limit arterial responses to the contractile influence of sympathetic nerves [13,14], a principal determinant of systemic blood pressure.

## 2. Postnatal Development Is Associated with Structural and Functional Remodeling of the Systemic Circulation

The mammalian cardiovascular system undergoes crucial structural and functional changes after birth. The arterial smooth muscle layer thickens and the amount of contractile proteins rises [15,16,17]. This increases the ability of arteries to develop a contractile response. In addition, the density of sympathetic innervation of the arteries increases gradually with age [15,18,19]. Therefore, arteries become more susceptible to procontractile influences from the sympathetic nervous system at adult age. Moreover, considerable alterations in the mechanisms governing vascular smooth muscle contraction occur with development—the strong contribution of mechanisms that are weakly dependent on changes in the intracellular calcium concentration (calcium sensitization) in the newborn switches to more calcium-dependent mechanisms as the organism grows [15,18,20,21]. Finally, the anticontractile influence of the vascular endothelium decreases with age [22,23]. This structural and functional remodeling leads to a dramatic rise of systemic blood pressure, which almost doubles from the time of birth [18,23,24,25].

The impact of such important regulators of vascular tone as potassium channels also changes during postnatal life. The present review focuses on the developmental alterations of the vasomotor role of different potassium channel families in arteries of the peripheral circulation, which play a key role in the regulation of systemic blood pressure.

## 3. Developmental Alterations of Potassium Channel Functioning in Arteries of the Systemic Circulation

### 3.1. Voltage-Gated Potassium Channels (K_v_ Channels)

#### 3.1.1. K_v_ Channels: Properties and Functions in Arteries

Voltage-gated potassium channels (K_v_ channels) are a large family of potassium channels which activate in response to membrane depolarization. A total of 12 subfamilies of K_v_ channels (K_v_1–K_v_12) exist, and the functionally most important subfamilies in arterial smooth muscle are K_v_1, K_v_2 and K_v_7 [26,27]. In arterial smooth muscle, the main isoforms of K_v_1 channels are K_v_1.2, K_v_1.5; of K_v_2 channels is K_v_2.1; and of K_v_7 channels are K_v_7.4 and K_v_7.5 [8,28]. A unique feature of K_v_7 channels is their relatively low activation threshold. The V_0_._5_ value (voltage that produces 50% activation) is +5 mV for K_v_1 and K_v_2, and only −34 mV for K_v_7, as was shown in the rat mesenteric artery smooth muscle cells [29]. In other words, the activation threshold of K_v_7 channels is close to the level of the resting membrane potential of arterial smooth muscle cells, which varies in the range from –40 to −65 mV [6,8,9]. A number of studies have demonstrated the important role of K_v_1, K_v_2 and, especially, K_v_7 channels in maintaining basal tone and regulating contractile responses of arteries, since the blockade of these channels caused depolarization of arterial smooth muscle cells and the development of strong basal tone and enhanced vasoconstrictor responses [26,30,31,32,33,34,35,36].

#### 3.1.2. The Vasomotor Role of K_v_ Channels Decreases with Maturation

In earlier studies, 4-aminopyridine (4-AP) was used as a blocker of K_v_ channels [37]. It was demonstrated that the cumulative addition of 4-AP caused contraction of isolated aorta segments from newborn rats, while its effects on adult aorta were considerably less pronounced [38]. In accordance with this, 4-AP inhibited [38] or completely blocked [39] the outward potassium current in isolated smooth muscle cells from newborn, but not adult, aorta. In contrast, the effects of 4-AP on resting tone and 5-HT (5-hydroxytryptamine or serotonin)-induced contractions were more pronounced in cerebral arteries of adult sheep than in fetuses [40], suggesting an increase in K_v_-functional contribution with maturation. However, in recent studies, 4-AP was shown to activate K_v_7.4 channels [41] and increase the intracellular pH affecting BK_Ca_ currents [42], which complicates the interpretation of the results obtained using this inhibitor.

Recently, selective blockers for K_v_1, K_v_2 and K_v_7 channels have become available: DPO-1, stromatoxin and XE991, respectively [33,43,44], which made it possible to evaluate the contribution of these K_v_ channel subfamilies to the regulation of arterial tone separately.

With the use of DPO-1 (1 µM), it was shown that the contribution of the K_v_1 subfamily decreases with maturation [45]. DPO-1 caused the development of basal tone, increased contractile responses and sensitivity to the α_1_-adrenoceptor agonist methoxamine in saphenous arteries of young rats (10 to 15 days old), while no effect was observed in arteries of adult animals (2 to 3 months old) [45]. The decrease in the functional impact of K_v_1 channels during development may be associated with a decrease in their expression. This assumption is supported by data showing a decrease in K_v_1.2 channel protein expression in rat aorta in the period between birth and adulthood [39].

Stromatoxin, a blocker of the K_v_2 subfamily, did not affect the contractile responses of saphenous arteries, either in young or in adult rats [45], although an increase in K_v_2.1 channel protein expression with maturation was previously demonstrated in rat aorta [39]. Perhaps the levels of expression and, consequently, functional contribution of K_v_2 channels may differ between the aorta and saphenous arteries.

The functional role of K_v_7 channels strongly declines with maturation. Blockade of these channels with XE991 and linopirdine led to the development of basal tone, increased contractile responses and sensitivity to the α_1_-adrenoreceptor agonist methoxamine and the thromboxane A2 receptor agonist U46619 in saphenous arteries of young rats [45,46]. These effects were considerably less pronounced in arteries of adult rats. The differences found were associated with a smaller effect of K_v_7 channel blockade on the membrane potential of smooth muscle in adult rats [45]. These effects are probably associated with a decrease in channel expression in arterial smooth muscle with development. Indeed, a decrease in K_v_7.4 channel protein expression was demonstrated with maturation. Importantly, the protein expression of the accessory KCNE4 subunit, which was shown to strongly augment K_v_7.4 channel protein expression and K_v_7.4 channel activity [47], also declined with age [45]. In addition to different expression levels, the functional impact of K_v_7 channels depends on the activity of other potassium channels. A recent study demonstrated that BK_Ca_ channel activity (which increases with age, see further in detail) suppresses K_v_7 channel functional availability in saphenous arteries of adult, but not young, rats [46].

Thus, the results with the use of selective K_v_ channel subfamily blockers allow the suggestion that the vasomotor role of K_v_1 and, especially, K_v_7 channels in systemic arteries decreases from the early postnatal period to adulthood. Moreover, their functional availability can be suppressed by the increasing contribution of other potassium channels, for example BK_Ca_ channels, with age.

### 3.2. Two-Pore Potassium Channels (K_2P_)

#### 3.2.1. K_2P_ Channels: Properties and Functions in Arteries

These channels were discovered in the vasculature relatively recently [48,49]. The channel is formed by two subunits, each has two pore-forming loops (P-loop, hence the name of the channel, K_2P_). Originally, K_2P_ channels were described as leakage channels, since at resting conditions they carry an outward potassium current, and the probability of channel opening does not depend on the membrane potential [11,50,51]. Research over the last few years has changed the image of these channels from being simply leak channels to regulatory channels having important roles in the control of cell excitability [52]. These facts indicate the involvement of K_2P_ channels in the maintenance of the resting potential, denoting their importance for the regulation of arterial tone.

There are six functional classes of K_2P_ channels. One of them, the acid-sensitive TASK-1 channel, is of particular interest. Recently, it was shown that a mutation in the *KCNK3* gene encoding the TASK-1 channel is associated with familial and idiopathic forms of pulmonary hypertension [53].

#### 3.2.2. The Impact of TASK-1 Channels on the Regulation of Vascular Tone and Blood Pressure Dramatically Decreases with Maturation

The functional role of these channels is important in pulmonary circulation [53,54,55], but very little is known about their function in systemic arteries. The ontogenetic aspect of TASK-1 channel function in systemic arteries has been studied very little. The only work devoted to this issue indicated a decrease in their vasomotor role in early postnatal ontogenesis [24].

Blockade of TASK-1 channels was shown to increase basal tone and contractile responses to methoxamine in the saphenous artery of young, but not adult, rats [24]. In arteries of adult rats, TASK-1 channel blockade did not lead to such a pronounced depolarization of the smooth muscle cells as was observed in young animals. Intravenous administration of AVE1231, a TASK-1 channel blocker [56], led to a considerable increase in mean blood pressure at young age, but not in adulthood, which indicates a decrease in the role of TASK-1 channels in blood pressure control with maturation. Finally, both TASK-1 mRNA and protein content in arterial tissue decreased as the rats grew older [24].

Therefore, the anticontractile role of TASK-1 channels in the arteries of systemic circulation and their impact to the lowering of blood pressure most likely declines with maturation.

### 3.3. Inward-Rectifier Potassium Channels (K_ir_ Channels)

#### 3.3.1. K_ir_ Channels: Properties and Functions in Arteries

Inward-rectifier potassium channels mediate an inward potassium current at membrane potentials more negative than the equilibrium potassium potential (E_M_ > E_K_) and a less pronounced outward potassium current at membrane potentials more positive than the equilibrium potassium potential (E_M_ < E_K_) [57]. The property of inward rectification is determined by the influence of intracellular substances—magnesium ions and polyamines, which block the outward potassium current at E_M_ < E_K_ [58,59]. There are seven subfamilies of K_ir_ channels, the second (K_ir_2) and sixth subfamilies (K_ir_6) are expressed in arterial smooth muscle [8]. The sixth subfamily of K_ir_ channels has distinct functional and regulatory features and therefore is classified into a class of ATP-dependent potassium channels (K_ATP_ channels) which will be discussed separately (see Section 3.4 below).

Arterial smooth muscle cells express K_ir_2.1, K_ir_2.2 and K_ir_2.4 isoforms. Using K_ir_2.1 gene knockout mice, it was demonstrated that this isoform is primarily functionally important in cerebral arteries at least in the early postnatal period, since the data were obtained on young animals [60]. When the level of the membrane potential of arterial smooth muscle is more positive than E_K_ (at physiological resting membrane potentials), K_ir_ channels mediate an outward hyperpolarizing current which counteracts the contraction of arteries. Indeed, a number of studies have demonstrated that K_ir_ channels are involved in maintaining the resting potential since their blocker barium chloride caused a depolarization of arterial smooth muscle [61,62,63]. Moreover, pretreatment of the rat coronary, cerebral and tail small arteries with barium chloride inhibited KCl-induced vasodilation, indicating that K_ir_2 channels mediate this reaction [61,63].

#### 3.3.2. The Vasomotor Role of K_ir_2 Channels Decreases with Maturation

The contribution of K_ir_2 channels on the regulation of vascular tone, as well as its expression pattern during ontogenesis, is not fully understood. It was shown that K_ir_2.1 and K_ir_2.4 mRNA expression decreased with age, while the expression of K_ir_2.2 increased during postnatal maturation in smooth muscle cells of the rat saphenous artery [45]. In functional experiments on the sheep middle cerebral artery, blockade of K_ir_2 channels with barium chloride did not affect basal tone and contractile responses to 5-HT in either adult sheep or sheep fetuses [40]. However, another agonist, noradrenaline, enhanced the contractile responses of middle cerebral arteries of sheep fetuses in the presence of barium chloride, which was not observed in arteries of adult animals [64]. In addition, barium chloride increased the contractile responses to the α_1_-adrenoceptor agonist methoxamine in rat saphenous arteries at different stages of development and the extent of this effect declined with age [45]. A greater contraction during the blockade of K_ir_ channels in young rats was accompanied by a greater depolarization of the smooth muscle membrane potential compared to adults [45].

Therefore, results of a few studies suggest a decrease in the functional role of K_ir_2 channels with age [45,64]. The decline of K_ir_2.1 and K_ir_2.4 expression may result in the decline of their vasomotor role. However, more studies on this issue should be conducted.

### 3.4. ATP-Sensitive Potassium Channels (K_ATP_ Channels)

#### 3.4.1. K_ATP_ Channels: Properties and Functions in Arteries

K_ATP_ channels are sensitive to the concentration of ATP in cell cytoplasm. The intracellular ATP concentration is quite high at physiological resting conditions and K_ATP_ channels are closed. However, under conditions of limited oxygen and substrates, ADP accumulates, which activates the K_ATP_ channel and leads to an efflux of potassium from the cell, hyperpolarizing the membrane potential. Thus, the ADP/ATP ratio is a key physiological regulator of K_ATP_ channel activity [65,66]. Relaxation of arteries under functional and reactive hyperemia occurs largely due to the activation of K_ATP_ channels [67,68]. It is important to note that under such pathological conditions as hypoxia, ischemia, and sepsis, the role of K_ATP_ channels in the regulation of arterial tone is extremely large [69,70].

#### 3.4.2. The Vasomotor Role of K_ATP_ Channels under Normal Physiological Conditions Does Not Change during Postnatal Development

The functional role of K_ATP_ channels during ontogenesis has been poorly studied. Of note, the results of these studies qualitatively depend on the method employed to assess the vasomotor role of K_ATP_ channels—using an activator or a blocker.

In one of the early studies, lemakalim, an activator of K_ATP_ channels, relaxed the middle cerebral artery in both adult sheep and fetuses [71]. At the same time, the sensitivity to lemakalim was considerably higher in adult sheep, which suggests an increase in the functional impact of K_ATP_ channels in the regulation of vascular tone with age. Later on, this group demonstrated a qualitatively similar result using another K_ATP_ channel activator—pinacidil. The incubation of segments of the middle cerebral artery with pinacidil resulted in a weakening of contractile responses to noradrenaline in adult sheep, but not in fetuses [64].

The K_ATP_ channel blocker, glibenclamide, did not increase the basal tone and contractile responses to noradrenaline of the middle cerebral artery in either adult sheep or fetuses [64]. Similarly, no effect of glibenclamide on basal tone and contractile responses to the α_1_-adrenoceptor agonist methoxamine was found in the rat saphenous artery at different stages of postnatal development [45].

Apparently, under normal physiological conditions, K_ATP_ channels are not active in the vasculature of either young or adult animals, and, therefore, the effect of their blockade on vascular tone cannot be observed. At the same time, results obtained with the use of activators suggest that the functional contribution of these channels grows as the organism ages. It is possible that the number of these channels increases with age, as it has been shown for other tissues, for example the rat myocardium [71].

### 3.5. Calcium-Activated Potassium Channels of Large Conductance (BK_Ca_ Channels)

#### 3.5.1. BK_Ca_ Channels: Properties and Functions in Arteries

Calcium-activated potassium channels of large conductance (BK_Ca_ channels) are regulated by both the membrane potential and the intracellular calcium concentration. The activity of BK_Ca_ channels increases with membrane depolarization with an increase in the intracellular calcium concentration [72].

An important physiological role of BK_Ca_ channels is that they provide a negative feedback regulation of arterial myogenic tone. It is known that ryanodine receptors (RyRs) of the sarcoplasmic reticulum adjoin to BK_Ca_ channels in the smooth muscle cell outer membrane [73,74]. Moreover, BK_Ca_ channels are co-localized with voltage-dependent calcium channels in the caveolae of arterial smooth muscle cells [75,76]. Thus, a local increase in the calcium concentration as a result of calcium sparks or activation of voltage-dependent calcium channels causes a short-term increase in the activity of BK_Ca_ channels, the efflux of potassium ions from the cell, hyperpolarization and, as a result, relaxation.

#### 3.5.2. The Vasomotor Role of BK_Ca_ Channels Increases during Postnatal Development

Most studies have demonstrated that the contribution of BK_Ca_ channels to the regulation of vascular tone in systemic circulation increases with age.

In rat cerebral arteries, the effects of BK_Ca_ channel blockade became more pronounced with age—the BK_Ca_ channel blocker, iberiotoxin, caused a contraction and an increase in the intracellular calcium concentration in arteries of adult (12–14 weeks old), but not newborn, rats (1–2 days old) [77]. Similarly, iberiotoxin led to an increase in basal tone and sensitivity to the α_1_-adrenoceptor agonist methoxamine in saphenous arteries of adult (2–3 months old), but not of young (10–15 day old), rats [45,46]. The effects of iberiotoxin on the contraction of sheep middle cerebral arteries also increased with age [40]. Later, this group also demonstrated an increase in the contractile responses of the carotid artery of adult sheep, but not fetuses, to 5-HT in the presence of iberiotoxin [78]. The effects of tetraethylammonium (TEA), another—albeit less selective—blocker of BK_Ca_ channels, also intensified with age: TEA caused more pronounced contractions of the aortic segments of 8- and 12-week-old rats compared to newborns and 4-week-old animals [38]. In addition, TEA enhanced contractile responses to the noradrenaline of the aorta of 8–12 week old, but not neonatal, rats [38]. These authors also demonstrated that the average density of the outward potassium current in isolated smooth muscle cells from the rat aorta increased with age. This was achieved by a considerable increase in the contribution of the calcium-dependent potassium current to the total outward potassium current between the age of 4 and 8 weeks [38]. In accordance with this, various blockers of BK_Ca_ channels (TEA, charybdotoxin and paxillin) had a suppressive effect on the outward potassium current in isolated smooth muscle cells from adult, but not newborn, aorta [38,39].

The more pronounced contribution of BK_Ca_ channels to the regulation of arterial tone in the adult organism may be due to several reasons. First, negative feedback regulation of vascular tone, as described above, requires co-localization of sarcoplasmic reticulum ryanodine receptors, voltage-gated calcium channels and BK_Ca_ channels in the plasma membrane. Although ryanodine receptors and BK_Ca_ channels are expressed in smooth muscle cells at newborn age, they are not yet organized into clusters; therefore, BK_Ca_ channels cannot function properly [77]. Indeed, a considerable part of calcium sparks does not activate BK_Ca_ channels in smooth muscle cells of newborn animals [79].

Second, the frequency of calcium sparks in the smooth muscle cells of newborn animals is considerably lower than in adults [77] and the amount of ryanodine receptors increases with age in, at least, the rat saphenous artery [80]. Probably, as a compensation for limited calcium availability, BK_Ca_ channels of immature arteries have a lower set point for calcium (the calcium concentration required for half-maximum activation at a membrane potential of 0 mV) as it was demonstrated in isolated smooth muscle cells of sheep cerebral arteries [81]. It was suggested that BK_Ca_ channels of fetal sheep arteries have a greater sensitivity to calcium because they are more phosphorylated by the protein kinase G [82,83].

Third, the expression of BK_Ca_ channels in smooth muscle cells can change as the organism matures. It was demonstrated that the mRNA expression of both the pore-forming α1- and the regulatory β1-subunits, where the latter has a positive effect on channel activity [84,85], increases with age [45,46]. In addition, the level of labeled charybdotoxin (a ligand of BK_Ca_ channels) binding was considerably lower in the aorta of 4-week-old compared to 12-week-old rats [38].

Finally, the biophysical properties of BK_Ca_ channels may vary at different developmental stages. BK_Ca_ channels of adult aortic smooth muscle cells remain open almost 10 times longer than the channels of fetal cells [86]. In addition, currents mediated by BK_Ca_ channels in isolated smooth muscle cells of the middle cerebral artery of adult female baboons activated faster at lower membrane potential values than in fetal cells [87]. Probably the differences in the biophysical properties of the BK_Ca_ channel between an adult and an immature organism correlate with the expression of the regulatory β1-subunit, which has a positive regulatory effect on BK_Ca_ channel function [84,85].

## 4. Conclusions

Taken together, the data described above demonstrate that the pattern of potassium channel expression and function in arterial smooth muscle changes dramatically while the organism matures (Figure 1, Table 1). Importantly, such alterations do not seem to be dependent on the embryological origin of the particular vascular bed, since similar patterns of changes were observed in the proximal part of the aorta and smaller arteries [88]. BK_Ca_ channels acquire an important role in the regulation of vascular tone with maturation, but their role in developing arteries is limited. BK_Ca_ channel activation by calcium sparks is weak in the newborn organism due to an unfinished co-localization process with RyRs of the sarcoplasmic reticulum and a low calcium spark frequency. Less calcium-dependent potassium channel types (K_v_1, K_v_7, TASK-1 and K_ir_2) play the main role in counteracting vasocontraction at early postnatal life. Indeed, contraction of the arterial smooth muscle depends much more on calcium in the adult age than in the early postnatal period. Therefore, while the vascular remodeling in postnatal development proceeds (including the formation of BK_Ca_/RyRs clusters), a switch of the potassium channels from less to more calcium-dependent ones occurs. Conditions affecting arterial smooth muscle potassium channels expressed at high levels during the early postnatal period, especially Kv7 and TASK-1 channels, may disturb the developmental switch of potassium channel contribution to arterial tone regulation. This may possibly lead to disturbances in potassium channel function in childhood. The latter in its turn may cause the development of cardiovascular disorders, including hypertension [3,89,90,91,92,93,94], the occurrence of which is steadily increasing in childhood [95].

## Figures and Tables

**Figure 1 ijms-22-05413-f001:**
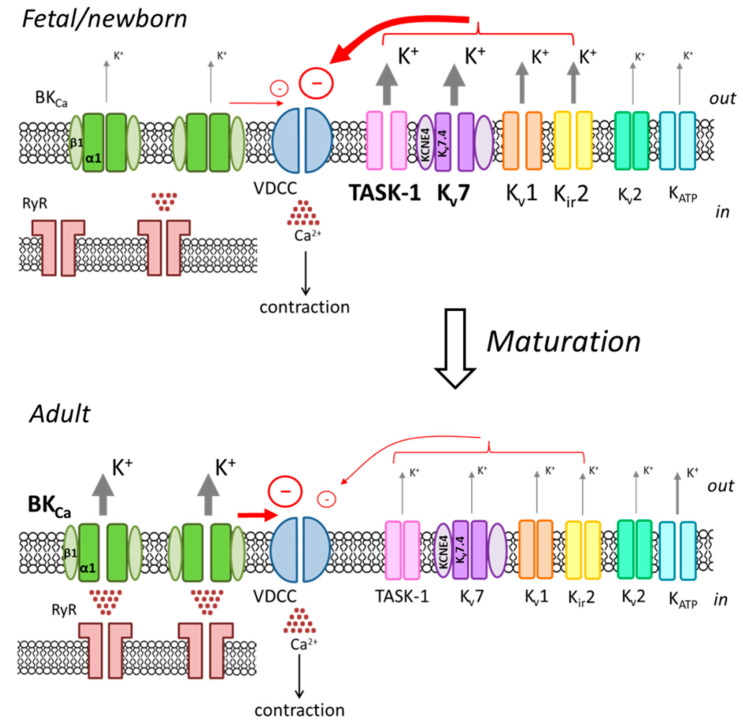
The switch of the leading role in the regulation of vascular tone from K_ir_2, K_v_1, K_v_7 and TASK-1 channels to BK_Ca_ channels with maturation. The activation of potassium channels leads to K^+^ efflux and hyperpolarization of the membrane, which in turn suppresses the opening of voltage-dependent calcium channels (VDCC) and counteracts vasocontraction. The arrow thickness represents the functional impact of the channel in this process.

**Table 1 ijms-22-05413-t001:** Alterations in the contribution of potassium channels to the regulation of vascular tone during maturation.

Channel Type		Alteration with Maturation	Objects Studied	Pharmacological Agent	References
K_v_	K_v_	Decrease	Rat aorta, segments and SMCs	4-AP (5 mM)	[38]
		Decrease	SMCs from rat aorta	4-AP (20 mM)	[39]
	K_v_1	Decrease	Rat saphenous artery	DPO-1 (1 µM)	[45]
	K_v_2	No change	Rat saphenous artery	ScTx (0.1 µM)	[45]
	K_v_7	Decrease	Rat saphenous artery	XE991 (3 µM), Linopirdine (10 µM)	[45]
		Decrease	Rat saphenous artery	XE991 (3 µM)	[46]
	K_v_	Increase	Sheep middle cerebral artery	4-AP (1 and 5 mM)	[40]
TASK-1		Decrease	Rat saphenous artery	AVE1231 (1 µM)	[24]
K_ir_	K_ir_2	Decrease	Sheep middle cerebral artery	BaCl_2_ (10 µM)	[64]
		Decrease	Rat saphenous artery	BaCl_2_ (30 µM)	[45]
		No change	Sheep middle cerebral artery	BaCl_2_ (10 µM)	[40]
	K_ir_6(K_ATP_)	No change	Sheep middle cerebral artery	Glib (0.1 mM–30 µM)	[64]
		No change	Rat saphenous artery	Glib (30 µM)	[45]
		Increase *	Sheep middle cerebral artery	Lemakalim * (0.01 nM–1 mM)	[71]
		Increase *	Sheep middle cerebral artery	Pinacidil * (1 nM–10 mM)	[64]
BK_Ca_		Increase	Rat cerebral arteries	IbTx (100 nM)	[77]
		Increase	Rat aorta, segments and SMCs	TEA (1–10 mM)	[38]
		Increase	Rat aortic SMCs	Paxilline (1 µM)	[39]
		Increase	Rat saphenous artery	IbTx (100 nM)	[45]
		Increase	Rat saphenous artery	IbTx (100 nM)	[46]
		Increase	Sheep middle cerebral artery	IbTx (100 nM)	[40]
		Increase	Sheep carotid artery	IbTx (100 nM)	[78]
		Decrease	SMCs from sheep cerebral arteries	IbTx (100 nM)	[81]
		No change	Sheep middle cerebral artery	IbTx (100 nM)	[64]

* An increase in K_ATP_ functional contribution was shown with the use of the activators lemakalim and pinacidil. All other substances listed above are blockers of potassium channels: 4-AP—4-aminopyridine; DPO-1—diphenyl phosphine oxide-1; Glib—glibenclamide; IbTx—iberiotoxin; ScTx—stromatoxin; TEA—tetraethylammonium.

## Data Availability

Not applicable.

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
