# Peer review of "Remodeling of Arterial Tone Regulation in Postnatal Development: Focus on Smooth Muscle Cell Potassium Channels"

_ijms, 2021, doi:10.3390/ijms22115413_

Round 1

Reviewer 1 Report

This review is interesting,  focusing on the effects of potassium channels on  vasomotor remodeling in postnatal development.

I have minor comments/suggestions.

  1. Consider to revise the title, such as vasomotor  or vascular tone remodeling.... based on the context of this review.
  2. Elaborate, in brief, how the sympathetic innervation impact on the activity of potassium channels, contributing to arterial contraction.
  3. The interaction of endothelial cells and smooth muscle cells play an important role in the regulation of vasomotor activity.  The potassium channels in endothelial cells may significantly influence on the tone of smooth muscle cells. It is interesting to be reviewed.
  4. it is known that pediatric hypertension is a problem and challenge in the modern society.  The disorder of potassium channels may play a causal/contributory role in the development of hypertension in the young.   It is very interesting to be discussed in this review.

Reviewer 2 Report

Vascular remodeling in postnatal development: focus on potassium channels

Dear Authors, I must congratulate team for their impressive work to review very complex but intriguing topic.

Manuscript is interesting and discussed very important topic of postnatal changes in various form of potassium channels. Manuscript well written but English editing is required.

Few question and points of discussion to author:

  1. I would like to check, what is clinical significance of potassium channel postnatal changes? It would be nice to have it as additional paragraph. This would potentially attract clinician to read and cite this manuscript.

  1. The VSMC has various way to control their tone. On my knowledge Ca2+ do play sone role in this process. Another important matter regarding VSMC related to VSMC plasticity with corresponding up and down regulation of contractile phenotype.  How does this process effect K+ channel?   This topic derive to be mention/discussed. (Role of Vascular Smooth Muscle Cell Plasticity and Interactions in Vessel Wall Inflammation by C. Shanahan. Frontiers in Immunology 11, 3053)

  1. Also, embryologically origin of arterial system comes from different origin. For example, ascending aorta and arch comes from neural crest it is different for descending aorta. Does it effect channel density or evolution? it should be discussed.  (Genetic and Epigenetic Mechanisms Underlying Vascular Smooth Muscle Cell Phenotypic Modulation in Abdominal Aortic Aneurysm. R Gurung, et al. International Journal of Molecular Sciences 21 (17), 6334)

Reviewer 3 Report

Here, Dr. Shvetsova and colleagues reviewed the developmental alterations of potassium channels during vascular maturation. They discussed the contribution shift from Kir2, Kv1, Kv7 and TASK-1 channels to BKCa channels with maturation and highlighted the potential importance of this matter in treating vascular diseases. In general, this is a good and insightful review article, which could potentially be of interest. However, I do have some suggestions to improve the clarity of the review:

  • Before entering section 1 about "Potassium channels are key regulators of arterial tone", I think the authors need to briefly introduce what is potassium channel (in general) and its role in cardiovascular system. Perhaps, it is also good to mention several (cardio)vascular diseases caused by potassium channel malfunction to stress that this is an impotant issue to be reviewed.
  • When explaining about the role of potassium channel in membrane potential (e.g., line 30), the authors need to describe its function in repolarization, in addition to the membrane stabilizing effect. Specifically, the word "repolarization" needs to be introduced.
  • Moreover, it will be useful to add an illustration of vascular smooth muscle (VSM) action potential (AP) traces describing the parts of AP where potassium channel functions and what happens to the AP when the channel malfunctions. This can be the new Figure 1.
  • Line 43: remove "normally", otherwise the authors are required to explain what happens in exceptional situations. 
  • Because the title does not explicitly mention arteries, please also discuss the contributions and the developmental changes of potassium channels in the veins (venous system). Is there any notable difference between arteries and veins? It would be insightful to have the comparison between the two, perhaps in a table.
  • Maybe "V0.5" as shown in line 76 would be better called "Vhalf" or "V50"? 
  • Please change the title of subsection 3.1.1 to somewhat like: "Kv channels: properties and functions in arteries smooth muscle action potential". I think it will make more sense because the discussed material in this subsection is about its role in membrane AP. Otherwise, add some information about the role of Kv channels beyond AP.
  • When explaining the findings of previous studies (e.g., lines 44-45, lines 80-82, lines 145-147, lines 184-186, etc.), try to briefly explain what they did (the methods) and specifically found in their studies. Otherwise, the paragraph is left hanging and the story becomes incomplete. 
  • In line 80, the authors said "a number of studies" but I only saw one reference attached to this sentence. Please revise either by adding more references or change into "a study". If the cited paper is a review, I would suggest to refer to the original articles in the review.
  • Line 84: Also, add a comma after "in earlier studies"
  • Line 90: add a comma after "[29]"
  • Line 91: change "later" to "in recent studies, ..."
  • Line 93: "with" should be "using"
  • Line 95: add a comma after "recently"
  • Line 135: add a citation to the sentence or remove it. The sentence does not add anything unless the authors cite the first paper demonstrating K2P in vasculature.
  • Lines 139-140: "research over last few years"
  • Line 143: change "...which means that they are important for..." into "..., denoting their importance for..."
  • Line 148: change "to" into "on"
  • Line 151 and 152: remove "-ing" from "functioning"
  • Line 153: "indicated"
  • Line 177: change "considered" to "discussed" and "chapter 3.4" to "section 3.4 below"
  • Consider replacing "has been studied very little" with "is not fully understood" or " has not been fully elucidated"
  • Line 192: add a comma after "artery"
  • I don't think I have seen the description about the 5-HT in the manuscript. Consider adding its function in potassium channel experiments.
  • Line 202: which studies? add the citation and explain the findings of these studies.
  • Line 188: change "to" into "on"
  • Line 211: remove "to" and change it into "...from the cell, hyperpolarizing the membrane potential."
  • Line 221: add a comma after "studies"
  • Line 224: maybe "Later on,..." is more appropriate?
  • Line 319: add a comma after "therefore"
  • Line 320: remove "process" and add a comma after closing bracket.
  • Table 1: change "pharmacological tool" to "pharmacological agent"
  • Please re-check if there are minor grammatical error, typo or commas required elsewhere.

Round 2

Reviewer 3 Report

Thank you for addressing my previous comments. I have no further issues.